# Towards Large Scale Transfer Learning for Differentially Private Image Classification

**Harsh Mehta**                     *harshm@google.com*
*Google Research*

**Abhradeep Thakurta**                 *athakurta@google.com*
*Google Research*

**Alexey Kurakin**                   *kurakin@google.com*
*Google Research*

**Ashok Cutkosky**                   *ashok@cutkosky.com*
*Boston University*

**Reviewed on OpenReview:** *https://openreview.net/forum?id=Uu8WwCFpQv*

## Abstract

Differentially Private Stochastic Gradient Descent (DP-SGD) has emerged as a popular private training algorithm. Unfortunately, the computational cost of training large-scale models with DP-SGD is substantially higher than non-private training. This is further exacerbated by the fact that increasing the number of parameters leads to larger degradation in utility with DP. In this work, we zoom in on the ImageNet dataset and demonstrate that, similar to the non-private case, pre-training over-parameterized models on a large public dataset can lead to substantial gains when the models are finetuned privately. Moreover, by systematically comparing private and non-private models across a range of large batch sizes, we find that similar to the non-private setting, the choice of optimizer can further improve performance substantially with DP. By using the LAMB optimizer, we saw improvement of up to 20% points (absolute). We also show that finetuning just the last layer for a *single step* in the full batch setting, combined with extremely small-scale (near-zero) initialization leads to both SOTA results of 81.7 % under a wide privacy budget range of $\varepsilon \in [4, 10]$ and $\delta = 10^{-6}$ while minimizing the computational overhead substantially. Finally, we present additional results on CIFAR-10 and CIFAR-100, surpassing previous state of the art by leveraging transfer learning with our recommendations.

## 1 Introduction

Despite significant research investment in privacy in machine learning, training private models remains challenging in practice. The standard metric for privacy is Differential Privacy (DP), which bounds the change in model weights (and predictions) if a single training example is added or removed. Informally, Differential Privacy implies that an adversary learns almost the same thing about an individual data point independent of their presence or absence in the data set, which intuitively provides protection against membership attacks (Nasr et al., 2021). Prior work illustrates that without DP's guarantees, it is quite plausible to attack a variety of models, across modalities, to reveal individual example information (Shokri et al., 2017; Carlini et al., 2019; 2021; Choquette-Choo et al., 2020; Liu et al., 2021; Balle et al., 2022). Revealing such information can be very dangerous if the model was trained using sensitive data such as demographics or personal photos, making it critical to establish practical DP training methods.

---

Code: https://github.com/google-research/google-research/tree/master/dp_transfer

Formally we can define DP as follows:

**Definition 1.1** (Differential Privacy (Dwork et al., 2006b;a))**.** *A randomized algorithm $\mathcal{A}$ is $(\varepsilon, \delta)$-differentially private if, for any pair of datasets $D$ and $D'$ differing in at most one example (called* neighboring datasets*), and for all events $\mathcal{S}$ in the output range of $\mathcal{A}$, we have*

$$\mathbf{Pr}[\mathcal{A}(D) \in \mathcal{S}] \leq e^{\varepsilon} \cdot \mathbf{Pr}[\mathcal{A}(D') \in \mathcal{S}] + \delta,$$

*where the probability is over the randomness of $\mathcal{A}$.*

The most popular method for DP training in deep learning is Differentially Private Stochastic Gradient Descent (DP-SGD) (Song et al., 2013; Bassily et al., 2014; Abadi et al., 2016). The core recipe implements a common theme in DP: "fuzzing" an algorithm's outputs with noise to obscure the contributions of any individual input. Specifically, DP-SGD is a preprocessing step prepended to ordinary SGD: for each example in a minibatch, compute the gradient, clip to a pre-selected norm, and add Gaussian noise before averaging over the minibatch and taking an SGD step.

In practice, DP training can be very expensive or even ineffective for very large models. A naive implementation might significantly increase the cost of bounding the sensitivity of each example as parameter count is increased. In addition, the norm of the noise added also increases, which is believed to lead to worse model quality. *Our goal is to develop simple and practical procedures for producing high-quality large-scale private models.*

To do this, we focus on *transfer learning* for improving the privacy-utility tradeoff in this large model regime. That is, we first pre-train a model on a large dataset with no privacy concerns, and only then privately finetune the model on the sensitive dataset. This strategy has emerged as a promising technique to improve the accuracy of private models. We specifically examine the effect of pre-training on JFT-300M, JFT-4B and ImageNet-21k datasets (Sun et al., 2017; Deng et al., 2009), and private finetuning on the popular ImageNet classification task with additional results on CIFAR-10 and CIFAR-100 datasets. Here, following Kurakin et al. (2022); De et al. (2022), we are simulating private training on a sensitive dataset by using the publicly available Imagenet data in place of the "sensitive" dataset. We show that using transfer learning, utility *improves* as the size of the model is increased, despite requiring more Gaussian noise. In addition, scaling both model and pre-training dataset size reduces the gap between private and non-private performance.

In search of further improvements, we recall that increasing batch size improves performance under DP (McMahan et al., 2018; Anil et al., 2021a; Kurakin et al., 2022; De et al., 2022). Thus, we consider very large batch sizes (within a small multiple of the full batch). With these large batch sizes, similar to the non-private setting, we find the choice of optimizer leads to substantial improvements when training the model with privacy. For instance, ResNet50x4 (BiT) obtains 68.8 % test accuracy when trained using DP-SGD with LAMB optimizer (denoted as DP-LAMB below) compared to 47.1% using DP-SGD with Momentum over a privacy guarantee of $(\varepsilon, \delta) = (10, 10^{-6})$.

Furthermore, we carefully explore other hyperparameters affecting DP performance. Surprisingly, we found the initialization of the last layer to be crucial: initializing the last layer to *zero* yields significant improvements. In summary, we investigate three distinct recommendations:

1. Increase pretraining data and model size.

2. Use large batch sizes in combination with optimizers that can work well with large batch sizes (e.g. LAMB).

3. Initialize the last layer to zero (or very small) when doing finetuning with DP.

Combining these recommendations, we achieve state-of-the-art DP results through finetuning just the last layer of ViT-H/14-4b. Remarkably, we only need to train for a *single step* in the full batch setting. Recently, Kurakin et al. (2022) obtain 47.9 % accuracy on a privacy budget of $(10, 10^{-6})$ with 70 epochs of finetuning and De et al. (2022) obtain an impressive result of 81.1% on privacy budget of $(8, 8 * 10^{-7})$ and 77.1% at $(1, 8 * 10^{-7})$, with 2500 epochs. Our results outperform the previous state of the art of Kurakin et al. (2022) and De et al. (2022) for all values of $\varepsilon$ considered in both studies. In addition, since our best results were

obtained by finetuning the last layer for a single epoch on ImageNet, our recommendations lead to at least 70x reduction in private training time than prior work and thus significantly improve the cost-utility ratio of training a high-quality image classification model with DP.

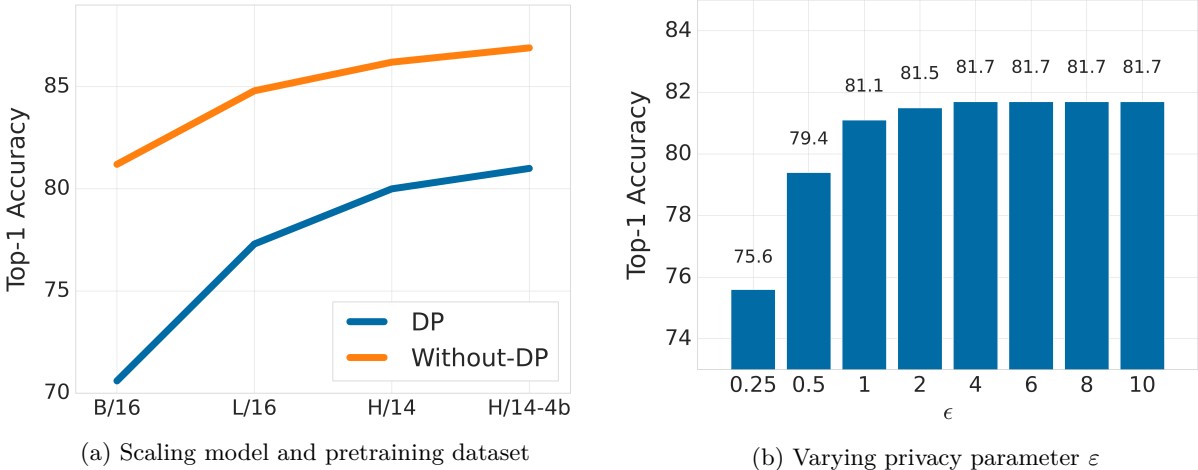

(a) Scaling model and pretraining dataset        (b) Varying privacy parameter $\varepsilon$

Figure 1: (a) Compares best models from our study with and without privacy on ImageNet across model and pre-training dataset sizes. Scaling the model (from ViT-B/16 → ViT-L/16 → ViT-H/14) and using a larger pre-training dataset (JFT-4b instead of JFT-300M) decreases the gap in accuracy coming from addition of the privacy guarantee of $(\varepsilon = 10, \delta = 10^{-6})$ (b) We finetune only the last layer of ViT-H/14-4b model for a *single step* in the full batch setting and vary $\varepsilon \in [0.25, 10]$. The previous state of the art of Kurakin et al. (2022) obtains 47.9 % accuracy on a privacy budget of $(10, 10^{-6})$ with 70 epochs of finetuning. More recently, concurrent work of De et al. (2022) obtain an impressive result of 81.1% on privacy budget of $(8, 8 * 10^{-7})$ and 77.1% at $(1, 8 * 10^{-7})$, with 2500 epochs. We present significant improvements on both studies at all $\varepsilon$ values we tried with only 1 epoch of finetuning. Note that these results are average over 5 independent runs with different seeds. For reproducability purposes, we also make the exact hyperparameters available in the Appendix.

## 2 Related Work

**Theoretical Understanding.** In the convex case, there is a rich literature on DP learning and optimization (Chaudhuri et al., 2011; Kifer et al., 2012; Song et al., 2013; Bassily et al., 2014; Talwar et al., 2015; Wu et al., 2016; Feldman et al., 2020a; Song et al., 2021; Asi et al., 2021), and the optimal privacy-utility tradeoffs are well-known. Such matching upper and lower bounds are not available in the non-convex context of large deep learning models. In practice, DP-SGD (Song et al., 2013; Bassily et al., 2014; Abadi et al., 2016) is the most popular learning algorithm. Although its optimality is not known, there is an active line of work on improving the theoretical properties of DP-SGD, including adaptive clipping (Andrew et al., 2021; Pichapati et al., 2019) and private aggregation of teacher ensembles (Papernot et al., 2018).

**Transfer Learning.** Several recent works demonstrate improved performance in the setting where we have access to a large public or non-sensitive dataset of the same modality as the private data (Tramèr & Boneh, 2021; Yu et al., 2021; Li et al., 2022; Kurakin et al., 2022; Hoory et al., 2021). Perhaps the direction most similar to ours is the concurrent work of De et al. (2022) which also employs JFT-300M as a public dataset to obtain high utility privately trained ImageNet models, corroborating some of the trends we observe as well. While the overall direction is similar, there are many notable differences including model families explored (ResNet (BiT) (Kolesnikov et al., 2020) and Vision Transformer (Dosovitskiy et al., 2021) vs NFResNet (Brock et al., 2021b)), batch size (full batch vs 16k) and choice of optimizers for best results (LAMB/ADAM vs Momentum).

**Large batch sizes.** In the context of differential privacy, several recent paper advocate the use of large batch sizes (McMahan et al., 2018; Anil et al., 2021b; Dormann et al., 2021; Hoory et al., 2021; Liu et al.,

2021; Kurakin et al., 2022) in order to improve the privacy-utility tradeoff. We confirm this finding in our setting as well, except for a few cases where the degradation in utility due to optimization difficulty with batch size increase trumped any potential increase in utility.

## 3 Background

We start our discussion with optimization details in the non-private setting. Given a data set $\mathcal{D} = \{(x_1, y_1), \cdots, (x_n, y_n)\}$ with $(x_i, y_i) \in \mathcal{D}$, we are interested in optimizing the function $\mathcal{L} : \mathbb{R}^d \times \mathcal{D} \to \mathbb{R}$ denoted as follows

$$\mathcal{L}(\theta) \triangleq \frac{1}{|\mathcal{D}|} \sum_{(x,y) \in \mathcal{D}} \ell(\theta, (x, y)) \tag{1}$$

The standard (non-private) approach to this problem is mini-batch Stochastic Gradient Descent (SGD). In a single iteration of SGD, a minibatch $B_t \subset \mathcal{D}$ of size $|B_t| = N$ examples are chosen randomly from the dataset. Then, given a current iterate $\theta_t$, SGD performs the update:

$$g_t \leftarrow \frac{1}{|B_t|} \sum_{(x,y) \in B_t} \nabla \ell(\theta_t; (x, y)) \qquad \theta_{t+1} = \theta_t - \eta_t g_t \tag{2}$$

where $\eta_t$ denotes the learning rate used for for iteration $t$.

DP-SGD is a private variant of this algorithm. In order to bound the sensitivity of each training example, Abadi et al. (2016) suggest computeing a gradient for each example separately and clip each a maximum norm of $C$ (a user-specified hyperparameter):

$$\tilde{g}_t \leftarrow \sum_{(x,y) \in B_t} \mathsf{clip}\left(\nabla \ell(\theta_t; (x, y))\right) \qquad g_t \leftarrow \frac{\tilde{g}_t + \mathcal{N}\left(0, \sigma C\right)}{|B_t|} \tag{3}$$

where $\mathsf{clip}(v) = v \cdot \min\left\{1, \frac{C}{\|v\|_2}\right\}$. This is notably different from non-private training where the forward pass can be vectorized and only a single pre-accumulated gradient need be calculated and used per mini-batch. If implemented naively, this step alone increases the computational cost of DP training proportional to the batch size for a single step. We do however note that, for several deep learning architectures, it is indeed possible to bound the sensitivity of each example without calculating the gradient for every example separately, leading to a dramatic cost-reduction both in terms of memory and compute (Goodfellow, 2015). One promising recent technique is Ghost Clipping (Li et al., 2022; Bu et al., 2022) where its possible to match the memory requirements of non-private training with marginal additional compute cost. In this paper, however, we stick to the exact scheme proposed in Abadi et al. (2016). Our recommendation for fine-tuning just the layer also reduces computational cost and can be used in conjunction with other cost-saving techniques like Ghost Clipping.

After summing the clipped example gradients, a noise vector sampled from a Gaussian distribution with standard deviation $\sigma C$ is added, where $\sigma$ is a parameter that will specify the privacy guarantee via the Gaussian mechanism. Formally, *any* algorithm that uses this noisy $g_t$ (not just SGD) will be private by the standard post-processing property of differential privacy.

## 4 Experiments

**Datasets.** We use the ILSVRC-2012 ImageNet dataset (Deng et al., 2009) with 1k classes and 1.3M images (we refer to it as ImageNet in what follows) as our final evaluation dataset. However, we provide supplementary results in Section F where evaluate on 2 additional datasets, namely CIFAR-10 and CIFAR-100. We also refer to these as the private dataset for which we want a privacy guarantee. We use 2 variants of JFT datasets for our pre-training as our public (in DP sense) dataset. JFT-300M (Sun et al., 2017) consists of 18k classes and 303M high-resolution images and JFT-4B with 29.5k classes and 4B images. Additionally, we also explore ImageNet-21k as our pretraining dataset in Section F.

**Public and private data.** We define "public data" and "private data" only in the context of DP. **Public data** is simply a dataset that we don't require privacy guarantee over and **Private data** is a sensitive dataset which we do want privacy guarantee over. "Public data" should not be confused with a dataset which is freely available. With this definition mind, it is completely plausible that a model trained with public data (in DP sense) still can not be shared widely without scrutiny due to either ethical, societal or legal concerns. For instance, the JFT datasets we use as "public data" are not technically publicly available but have been used extensively as a pre-training dataset in the non-private setting to obtain state-of-the-art results (Dosovitskiy et al., 2021; Brock et al., 2021a; Tolstikhin et al., 2021; Zhai et al., 2021).

Finally, note that none of our finetuning datasets (e.g. ImageNet-1K) in reality are sensitive datasets: we are only simulating a public/private dataset split only for demonstration purposes (Kurakin et al., 2022; De et al., 2022). To make sure that our simulated "public" and "private" datasets capture a practical scenario, we carefully de-duplicate our pre-training datasets w.r.t. **all** splits of our finetuning datasets (Kolesnikov et al., 2020; Dosovitskiy et al., 2021). More details about this process can be found in the appendix.

**Model variants.** We evaluate the transfer learning capabilities of ResNet (He et al., 2016) and Vision Transformer (ViT) (Dosovitskiy et al., 2021) in our study. Note that in typical non-private training, ResNet employs batch normalization, which has a difficult-to-analyze effect on the sensitivity of the gradients and so therefore makes computing the privacy parameters $\varepsilon$ and $\delta$ difficult. Somewhat fortuitously, to improve transfer, Kolesnikov et al. (2020) replace Batch Normalization with Group Normalization and used standardized convolutions, which has a much simpler privacy analysis. We therefore follow the same practice and denote our modified model "Resnet (BiT)". For Vision Transformer, we follow the standard notation to indicate the model size and the input patch size, for example, ViT-B/32 means the "Base" variant with 32x32 input patch size. Note that for ViT, compute requirements scales up as we reduce the patch size.

**Training details.** At the pre-training stage, we stick with the common practice of employing Adam optimizer (even for ResNet) (Kingma & Ba, 2014) with $\beta_1 = 0.9$ and $\beta_2 = 0.999$, with a batch size of 4096 and high weight decay of 0.1 unless mentioned otherwise. We train with sigmoid cross-entropy loss and use linear learning rate warmup until 10k steps, followed by linear decay until the end of training. For our private finetuning experiments, we stick with a reasonably stringent privacy guarantee of $\varepsilon = 10$ and $\delta = 10^{-6}$, unless specified otherwise. We use DP-SGD privacy analysis to compute the noise multiplier. To limit other confounding factors we set the clipping norm $C$ to 1. Also, since training with DP-SGD is computationally expensive, we finetune on ImageNet for at most 10 epochs. Finally, when training the last layer with DP we found it to be crucial to initialize the last layer weights to zero (or a small value). We expand on this more in Section 5.

## 4.1 DP-SGD with Large Batch Sizes

There is an inherent tradeoff between privacy enforced by DP and utility obtained from the model when trained privately. The decrease in the quality of the model stems from the additional randomness added to the gradients as part of DP-SGD. Even though there is precedence for gradient noise leading to superior generalization ability (Neelakantan et al., 2015; Smith et al., 2020), more often than not, in the context of differential privacy it leads to substantial degradation in model quality (Song et al., 2013; Bassily et al., 2014; Abadi et al., 2016).

DP-SGD first adds per-example gradients in a mini-batch, then adds a noise vector to the accumulated gradient, and finally divides by the batch size. Thus, increasing the batch size might seem to decrease the noise added as $\frac{\sigma}{|B_t|}$ decreases. The picture is actually more complicated: increasing batch size also decreases the effect of amplification by sampling, which means that the increase in $|B_t|$ must be partially offset by an increase in $\sigma$ in order to maintain the same privacy guarantee. Regardless, even for large models, others (McMahan et al., 2018; Anil et al., 2021b; Dormann et al., 2021; Hoory et al., 2021; Liu et al., 2021; Kurakin et al., 2022) have observed that employing larger batch size improves performance of DP-SGD. In our case, ImageNet contains roughly 1.3M examples; so we perform finetuning in the range of 128k ($2^{17}$) - 1M ($2^{20}$) batch size across our exploration and use the full batch setting for the final results.

| | | Without privacy | | | | With privacy | | | |
|---|---|---|---|---|---|---|---|---|---|
| | | Batch Size | | | | | | | |
| Model | Tuning type | $2^{17}$ | $2^{18}$ | $2^{19}$ | $2^{20}$ | $2^{17}$ | $2^{18}$ | $2^{19}$ | $2^{20}$ |
| ViT-B/16 | Full | 75.8 | 69.5 | 57.9 | 57.0 | 66.5 | 57.7 | 57.8 | 57.9 |
| | Last layer | 81.1 | 80.2 | 75.1 | 69.2 | 61.3 | 65.1 | 66.6 | 65.3 |
| ResNet50x4 (BiT) | Last layer | 74.1 | 69.4 | 61.9 | 46.1 | 39.6 | 43.6 | 47.1 | 49.2 |

Table 1: Comparison of Top-1 test accuracies on Imagenet when trained using DP-SGD with Momentum when using various models with and without privacy. All models are pretrained on JFT-300M dataset. Note that we exclude ResNet50x4 full finetuning results since we did not see accuracies greater than 20% in in our hyperparameter range even without privacy.

We finetune on ImageNet using 2 different schemes: 1) Full transfer learning, where we finetune all the parameters of the model and 2) Last layer tuning, where we only train a classifier produced from the features obtained from the pre-trained model. Despite all the recent advances made in engineering efficient implementations of DP-SGD for large models, last layer tuning is significantly more cost-effective in the context of DP-SGD since per-example gradients are trivial to compute and don't lead to much additional compute cost in comparison to non-private training. In fact, last layer tuning can also be cast as a linear regression problem and solved exactly, but in this study, we stick with gradient-based optimization methods which let us rely on DP-SGD's privacy accounting for all our experiments. Finally, in both types of finetuning methods, we reinitialize the last layer and train it from scratch.

As the first step in this section, we start with pre-training on 2 medium-sized models, namely 1) ViT-B/16 and 2) ResNet50x4. We conduct finetuning in almost the same fashion as reported in non-private cases except our batch sizes are much larger and we replace the traditional Momentum Optimizer with DP-SGD with Momentum.

As shown in Table 1, as the batch size increases, performance in the non-private case decreases significantly as optimization quality degrades. However, with DP, we sometimes observe the reverse trend, even to the point where it is *better* than non-private training (ResNet50x4, batch size $2^{20}$). Perhaps surprisingly, our results also suggest that training the full model is not necessarily better than training just the last layer. This is true even without privacy. We conjecture that this phenomenon is due to the optimization difficulty with large batch sizes.

## 4.2 Optimization Difficulty with Large Batch Sizes

Our results in Table 1 show that increasing the batch size improves the privacy-utility tradeoff (except for ViT-B/16-Full). However, without privacy, the performance seems to suffer as the batch size is increased in all cases. This phenomenon, outside the context of differential privacy, has been studied by others already (Krizhevsky, 2014; Li et al., 2014; Goyal et al., 2017; You et al., 2017; Hoffer et al., 2018). Most notably, You et al. (2019) introduced LAMB optimizer to address this issue, which modifies AdamW (Loshchilov & Hutter, 2017) with a layer-wise adaptive learning rate correction. With LAMB, they trained both ResNet-50 and BERT with batch sizes up to 32k. In this work, we would like to train our models with even larger batch sizes i.e. in the range of 128k-1M. Tuning only the learning rate, in our results in Table 2, we extend the use LAMB optimizer for both models in this batch size range, with and without privacy on ImageNet.

Switching from Momentum to LAMB provides improvements in several dimensions. First, without privacy, performance at the lower end of our batch size range comes quite close to that when trained with a much lower batch size of 512 (as proposed by Dosovitskiy et al. (2021)). Most notably, we see a big jump in performance for ResNet-50x4 when training the full model. Test accuracy on ResNet-50x4 with momentum did not surpass 20% (and hence was excluded from Table 1). In contrast, test accuracy with LAMB was comparable to ViT-B/16 across batch sizes. Second, the batch size most attractive in our context is $2^{20}$ (around 1M). Note this batch size is close to the ImageNet training dataset size of around 1.28M. We observe a

|       |              |      | Without privacy | | | | With privacy | | | |
|-------|--------------|------|----------|----------|----------|----------|----------|----------|----------|----------|
|       |              |      | Batch Size | | | | | | | |
| Model | Tuning type  | Best | $2^{17}$ | $2^{18}$ | $2^{19}$ | $2^{20}$ | $2^{17}$ | $2^{18}$ | $2^{19}$ | $2^{20}$ |
| ViT-B/16 | Full      | 84.8 | 81.7 | 77.1 | 76.3 | 73.1 | 68.2 | 70.5 | 70.5 | 71.1 |
|          | Last layer | 81.2 | 80.7 | 76.4 | 75.9 | 73.3 | 66.4 | 69.1 | 70.1 | 70.6 |
| ResNet50x4 | Full    | 85.3 | 83.7 | 79.5 | 74.5 | 72.6 | 55.6 | 64.5 | 68.6 | 70.1 |
|            | Last layer | 82.9 | 82.7 | 79.1 | 73.5 | 72.7 | 52.5 | 61.7 | 66.1 | 68.8 |

Table 2: Comparison of Top-1 test accuracies on Imagenet when trained using DP-LAMB when using various models with and without privacy. Similar to Table 1, all models are pretrained on JFT-300M dataset. The numbers in Best column are obtained by using Momentum Optimizer in non-private setting with batch size 512 and rest of the hyperparams reproduced from Dosovitskiy et al. (2021)

|       |              |      | Without privacy | With privacy |
|-------|--------------|------|-----------------|--------------|
|       |              |      | Batch Size | |
| Model | Tuning type  | Best | $2^{20}$ | $2^{20}$ |
| ResNet152x4 (BiT) | Full      | 87.1 | 77.0 | 72.9 |
|                   | Last layer | 84.6 | 76.9 | 72.9 |
| ViT-L/16          | Full      | 87.2 | 79.6 | 77.4 |
|                   | Last layer | 84.8 | 79.4 | 77.3 |
| ViT-H/14          | Full      | 88.2 | 81.5 | 80.1 |
|                   | Last layer | 86.2 | 81.4 | 80.0 |
| ViT-H/14 - 4B     | Full      | 88.7 | 82.5 | 81.0 |
|                   | Last layer | 86.9 | 82.5 | 81.0 |

Table 3: Comparison of Top-1 test accuracies on Imagenet when the model and dataset size is increased. We observe consistent improvement with scaling model from ViT-B/16 to ViT-L/16 and even ViT-H/14. Similarly, scaling the pretraining dataset from JFT-300M to JFT-4B also improves both private and non-private performance. Similar to before, all models are trained using DP-LAMB with and without privacy. The numbers in Best column are obtained by using Momentum Optimizer in non-private setting with batch size 512 and rest of the hyperparams reproduced from Dosovitskiy et al. (2021)

significant increase in accuracy when switching to LAMB for batch size $2^{20}$. Lastly, the increase in accuracies in the non-private setting helps when training the models privately across the board, to the extent that we obtain the best private finetuning numbers at the largest batch size of $2^{20}$.

## 4.3 Scaling Analysis

In this section, we systematically study the effect of scaling both the model size and the pre-training dataset. We follow Dosovitskiy et al. (2021) and experiment with 3 additional models, namely ResNet152x4, ViT-L/16 and ViT-H/14. Similar to earlier experiments, we pretrain these models with JFT-300M. To study the effect of increasing pre-training data, we consider one more variant "ViT-H/14 - 4b" where we pre-train a ViT-H/14 model on JFT-4B dataset for a single epoch. Due to the excessive computational cost of running these experiments, we only consider a batch size of $2^{20}$ for all our results in this section.

As shown in Table 3, for both the model families, namely ResNet (BiT) and Vision Transformers, increasing the model size improves accuracy in both non-private and private settings. We find this to be the case regardless of whether we train the full model or just the last layer. Perhaps even more encouraging is the fact that the gap between the performance with privacy and best numbers without privacy also decreases as the model size increases. For instance, the gap for ViT-L/16 (last layer tuning) between the best non-private and private case is 7.5 percentage points and for ViT-H/14 in the same setting is 6.2 percentage points.

We also present results when the pre-training dataset size is increased when we train using a much larger JFT-4B dataset instead of JFT-300M. Note that in this case we only train for a single epoch with JFT-4B which results in a similar number of pre-training steps if we were to train with JFT-300M for 14 epochs. This means that changes in performance between ViT-H/14 and ViT-H/14-4b can be largely attributed to the diversity of examples seen and not necessarily "more" pre-training. As shown in the best column in Table 3, we notice moderate improvement (about half a percentage point) between ViT-H/14 and ViT-H/14-4b but when training with privacy at a much larger batch size we see an even larger improvement (a full percentage point).

Lastly, when comparing full finetuning and just last layer tuning on ViT-H/14-4b setting, we still see a noticeable gap in the best column (trained with 512 batch size). However, when trained at a much larger batch size of $2^{20}$, the difference shrinks significantly regardless of whether it was trained with privacy or not. This is quite fortuitous since the computational cost of training with privacy is much larger when we finetune the full model vs finetuning just the last layer. In the former case, vanilla DP-SGD finetuning as proposed in Abadi et al. (2016) needs access to the per example gradient of the full model whereas when tuning only the last layer, the per example gradient of just the last layer would suffice. The difference can be even more striking when the models are made larger (Appendix Table 10). Note that when tuning only the last layer we still perform a forward pass again every time an example is visited largely due to data augmentation. An alternative is to perform the forward pass for the whole dataset once and cache to features to be trained with DP. This is further simplified when data augmentation is turned off.

## 5 Influence of other hyperparameters

Across our exploration so far, the only hyperparameter we tuned was the learning rate. We did this to limit the scope of our exploration since each added variable would result in a multiplicative increase in compute.

Outside the context of differential privacy though, good performance on image classification task can depend on a lot of other factors, such as data augmentation (Perez & Wang, 2017; Cubuk et al., 2018), initialization (Hanin & Rolnick, 2018; Zhang et al., 2019; Mehta et al., 2021; Brock et al., 2021c), feature/data normalization (Ioffe & Szegedy, 2015; Wu & He, 2018), resolution of the input image, training iterations etc. It is possible that when trained with privacy, the optimal choices of these settings are different. Thus, in this section, we systematically unpack a potentially mysterious bag of choices when training the model with a privacy guarantee. To do this, we take the best performing model so far i.e. ViT-H/14 - 4b, and change one hyperparameter at a time as shown in Table 4. Note that since there was no performance difference between training the full model vs last layer tuning on ViT-H/14 - 4b (Table 3), we only tune the last layer for this study.

| ViT-H/14 - 4b | Accuracy | Δ |
|---|---|---|
| No change | 81.0 | – |
| DP-LAMB → DP-Adam | 81.0 | – |
| DP-LAMB → DP-SGD with Momentum | 77.2 | -3.8 |
| Zero Init → Lecun Normal Init | Random Chance | -81.0 |
| Inception Crop → Center Crop | 81.2 | +0.2 |
| 10 epochs → 1 epochs | 81.1 | +0.1 |
| Clip at 1 → 10 | 80.1 | -0.9 |
| Clip at 1 → 0.1 | 81.0 | – |
| Resolution 384 → 512 | 79.0 | -2.0 |
| Resolution 384 → 256 | 81.4 | +0.4 |

Table 4: Taking the best performing model from the previous sections (ViT-H/14-4b), we systematically study the effect of changing a single hyperparameter at a time. Combining the positive deltas from this study led to a cumulative improvement and we used the resulting model to present the numbers in Figure 1b.

**Choice of optimizer.** As shown in Table 4, switching from DP-LAMB to DP-Adam did not change at all in this setting, however, switching to DP-SGD with Momentum leads to a noticeable decrease in accuracy. This suggests that the layer-wise adaptation of the learning rate, as suggested by LAMB, is much less crucial than the contribution made by the Adam update. This is perhaps not surprising since we are only learning the last layer and the layerwise adaptivity provided by LAMB can be subsumed in the learning rate, which we tune.

**Initialization.** From our results, the choice of initialization seems to be a crucial hyperparameter. Switching the last layer from Zero Initialization to Lecun Normal (default in Flax for dense layers) resulted in a complete lack of progress and random chance performance on the test set when we trained privately. In the non-private case, Zero initialization has a special property that the learning rate used *in the first step* can be decreased arbitrarily (up to precision limits) since it only affects how much the loss decreases and not the accuracy (since we use sigmoid cross-entropy loss). This is useful when training privately since the update is artificially made noisy for the privacy guarantee. In this case, the magnitude of the update can be controlled using the learning rate without changing the potential accuracy obtained by the original gradient before the noise was added. This argument is very similar in spirit to the one made by Bu et al. (2021) who show that noise multiplier $\sigma$ vanishes and doesn't affect the convergence in the gradient flow regime. Note that there is nothing special about initializing at absolute zero. As long as the **initialization is small enough** to allow setting small learning in the first step, we find, also leads to similar results as shown in Figure 2c.

**Number of epochs.** Since every visitation of the dataset affects the privacy budget, training for less number of epochs requires a lower noise multiplier $\sigma$ for the same values of $(\varepsilon, \delta)$. Typically, the advantage with less noise is trumped by the gain in performance by training longer (Kurakin et al., 2022; De et al., 2022). However, we observe that training for a single epoch performs slightly better than training for 10, as shown in Table 4.

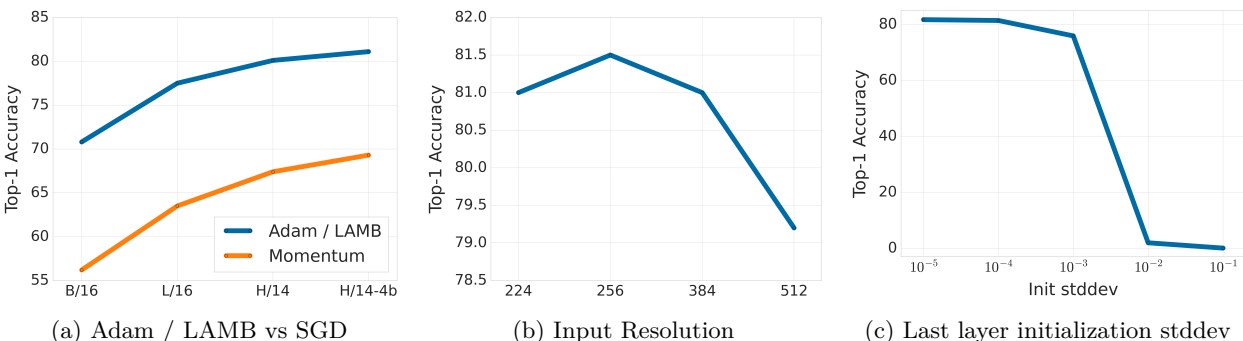

| (a) Adam / LAMB vs SGD | (b) Input Resolution | (c) Last layer initialization stddev |

Figure 2: We consider ViT-H/14-4b model in the single step, full batch setting where only the last layer is finetuned and plot a more fine-grained relationship between a few important hyperparameters and test accuracy on ImageNet. (a) LAMB outperforms SGD with Momentum in non-private setting when only the last layer is finetuned. Note that in the first step since the norm of the weights is zero, LAMB is equivalent to Adam. (b) We find a non-trivial relationship between input resolution and large batch training where a sweet spot of 256x256 image resolution performs best. (c) Increase in scale of initialization of the last layer degrades test accuracy when trained with DP. With big enough initialization, test accuracy degenerates to random chance performance. Note that the phenomenon observed in (a) and (b) can be attributed to the use of large batch sizes and not necessarily DP but (c) is more tied to the use of DP, as larger initializations are actually *standard* without DP.

## 5.1 Extreme case of full batch and single step update

Since we observed best results when the model was trained for a single epoch, we consider an extreme case where we finetune the model for a **single step** in the full batch setting. We also combine all the positive changes from Table 4, namely 1) replacing Inception Crop with Central Crop and 2) changing the resolution of the input image from 384 to 256. As shown in Figure 1b, as we hoped, these changes provide improvements

| Finetuning Dataset | Non-Private | Epsilon | | | | |
|---|---|---|---|---|---|---|
| | | 0.5 | 1.0 | 2.0 | 4.0 | 8.0 |
| CIFAR-10 | 95.2 | 95.1 | 95.1 | 95.1 | 95.1 | 95.2 |
| CIFAR-100 | 79.9 | 78.3 | 79.5 | 79.7 | 79.7 | 79.7 |
| ImageNet-1k | 73.8 | 19.0 | 39.1 | 54.2 | 63.3 | 68.3 |

Table 5: Comparison of Top-1 test set accuracies on when pre-training ViT-B/16 model on ImageNet-21k dataset. All other details are present in the Appendix Section F.

almost orthogonal to each other, through which we obtain SOTA results on ImageNet across all values of $\varepsilon$ we tried. Finally, we also observe consistent improvements from just the scale of pre-training in Figure 1a.

### 5.2 Pretraining with ImageNet-21k

In this section, to test our own recommendations on other tasks, we explore using ImageNet-21k as our pre-training dataset. As shown in Table 5, even when pretrained on much smaller ImageNet-21k dataset, our recommendations lead to beating previous state of the art of 47.9% accuracy on ImageNet-1K Kurakin et al. (2022) with only 1 epoch of finetuning on last layer features. For CIFAR-10, (Klause et al., 2022) obtain an impressive 82.5% top-1 accuracy for $\varepsilon = 8$ without leveraging transfer learning. With private finetuning, we improve this result to 95.2% at $\varepsilon = 8$. Finally, at stringent epsilon values like $\varepsilon = 1$, our results outperform best results of 94.7% on CIFAR-10 and 70.3% on CIFAR-100 from concurrent work of De et al. (2022) even when additional data is used (i.e. employing transfer learning).

## 6 Conclusion

We demonstrate that large-scale pretraining on public dataset is an effective strategy for obtaining good results when finetuned privately. Moreover, scaling both model size and pre-training dataset improves performance of the private model and closes the gap between the accuracy obtained in the non-private setting. We also find that gains from transfer learning can be further amplified by switching to an optimizer that works well with large batch sizes, LAMB in our case. In addition, our exploration allowed us to obtain a significant improvement over existing results on training ImageNet privately a wide range of $\varepsilon$ by finetuning just the last layer for only 1 epoch (1 step in full batch setting as shown in Figure 1b). This significantly reduces the computational cost of training with privacy. Additionally, recent low-rank update techniques other than last layer finetuning can also be leveraged for effective DP training (Yu et al., 2021). We leave this interesting direction to future work. Finally, similar to the baselines, we note that our privacy accounting do not include any cost of hyperparameter tuning, which is another direction of interesting subsequent exploration (Papernot & Steinke, 2022).

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

## A Algorithmic details

We present below a generalized version of DP-SGD where the gradients get processed in the traditional DP-SGD fashion and are then passed to a first order optimizer as an input. This lets us instantiate DP versions of well known optimizers like SGD, Momentum, Adam and LAMB. We prepend the optimizer's name with **DP** to denote that the gradients were first processed as shown in Algorithm 1 and then passed to the said optimizer.

---

**Algorithm 1** Generalized First Order Differentially Private Algorithm

---

**Require:** Data set $D = \{(x_1, y_1), \cdots, (x_n, y_n)\}$ with $(x_i, y_i) \in \mathcal{D}$, loss function: $\ell : \mathbb{R}^d \times \mathcal{D} \to \mathbb{R}$, a first order optimizer $Opt$, clipping norm: $C$, number of iterations: $T$, noise multiplier: $\sigma$

1: Randomly initialize $\theta_0$.
2: **for** $t = 0, \ldots, T - 1$ **do**
3:     Select randomly without replacement a mini-batch of examples $B_t \subseteq D$
4:     $g_t \leftarrow \sum_{(x,y) \in B_t} \mathsf{clip}\left(\nabla \ell(\theta_t; (x, y))\right)$, where $\mathsf{clip}(v) = v \cdot \min\left\{1, \frac{C}{\|v\|_2}\right\}$.
5:     $\tilde{g}_t \leftarrow \frac{1}{\|B_t\|} g_t + \mathcal{N}\left(0, (\sigma C)^2\right)$
6:     $\theta_{t+1} \leftarrow$ single step of first order optimization with gradient $Opt(\tilde{g}_t)$
7: **end for**
8: **return** $\frac{1}{T} \sum_{t=1}^{T} \theta_t$ or $\theta_T$.

---

## B Privacy Analysis

The privacy parameters $(\varepsilon, \delta)$ are functions of $C$, $\sigma$, $|B_t|$, $|\mathcal{D}|$, and the total number of iterations $T$. DP-SGD algorithm involves setting the right clipping norm $C$ and the noise multiplier $\sigma$ given a privacy budget, batch and dataset size. The $(\varepsilon, \delta)$ guarantee is computed by analysis of the Gaussian Mechanism with privacy amplification by subsampling and composition across across iterations (Kasiviswanathan et al., 2008; Bassily et al., 2014; Abadi et al., 2016; Mironov, 2017; McMahan et al., 2017; Mironov et al., 2019; Erlingsson et al., 2019; Zhu & Wang, 2019; Feldman et al., 2020b; Wang et al., 2020). Our implementation relies on Tensorflow Privacy [1] codebase for conversion of $(\varepsilon, \delta)$ and clipping norm $C$ to/from noise multiplier $\sigma$.

To put the epsilon-delta values in context, our best model obtains 81.7% accuracy for $\varepsilon \in [4, 10]$. Privacy guarantee for $\varepsilon \approx 4$ (Figure 1a) in fact satisfy a much stronger property of zCDP $<=1$ (0.154 for $\varepsilon = 4$) which is by now an industry standard.

## C Training details

The core bottleneck of DP-SGD is the computational cost of computing per-example gradients. A naive implementation of per-example gradient calculation can lead to a dramatic reduction of throughput and an increase in memory usage (proportional to the batch size) compared to non-private training. Inspired by Subramani et al. (2020), we conduct all our experiments in Jax (Bradbury et al., 2018; Frostig et al., 2018) is framework that leverages just-in-time compilation using XLA[2] and does auto-vectorization of the backward pass. We leverage this functionality throughout our experiments. Finally, we conduct our experiments on TPUv4 architecture.

We conduct all our experiment using Scenic library (Dehghani et al., 2021) for high quality reproducible implementations of both ResNet (BiT) and Vision Transformers. Scenic, in turn, uses Flax (Heek et al., 2020) for may of the layer definitions. We will also open-source our implementation of DP-SGD and DP-LAMB which includes vectorized per-example gradient clipping in a distributed training setup for further auditing by the research community. For the privacy accounting, we rely on the default Rényi accountant implementation already open-sourced as part of Tensorflow Privacy library.

---

[1] https://github.com/tensorflow/privacy
[2] https://www.tensorflow.org/xla

### C.1 Model details

We use 2 competitive model families for our experiments. 1) ResNet (BiT) and 2) Vision Transformer.

**ResNet (BiT).** Originally proposed in (Kolesnikov et al., 2020), it modifies the original ResNet architecture (He et al., 2016) by replacing Batch Normalization with Group Normalization (Wu & He, 2018) and additionally using Weight Standardization for the convolutional layers (Qiao et al., 2019).

**Vision Transformers.** We use the exact architecture and notation proposed by (Dosovitskiy et al., 2021). Vision Transformer creates 2d input patches of images and applies the Transformer backbone typically used NLP tasks. In addition to the models trained in the original paper, we make one addition of ViT-H/14-4b, denoting ViT-H/14 size model pretrained on the larger JFT-4B dataset.

## D  Pretraining details with JFT

**Datasets.** We use 2 variants of JFT datasets for our pre-training. JFT-300M (Sun et al., 2017) consists of 18k classes and 303M high-resolution images, while JFT-4B consists of 29.5k classes with 4B images. Note that we really do mean JFT-4B, which is a larger version of JFT-3B as used in Zhai et al. (2021).

**Deduplication.** In order to both not inflate our results and break privacy guarantee offered by finetuning privately on ImageNet, we extend the deduplication process proposed by Kolesnikov et al. (2020) and deduplicate both JFT-300M and JFT-4B with respect to all splits of ImageNet. We use a model based deduplication system which removes both exact and near-duplicates across common image transformation like crop, shift, resize etc.

**Hyperparameters.** At the pre-training stage, we stick with the common practice of employing Adam optimizer (even for ResNet) with $\beta_1 = 0.9$ and $\beta_2 = 0.999$, with a batch size of 4096 and high weight decay of 0.1. We also use linear learning rate warmup until 10k steps and linearly decay it until the end. All our models are pre-trained with 224x224-sized JFT images.

| Model | Epochs | Base LR | TPU v4 hours | Model size |
|---|---|---|---|---|
| ViT-B/16 | 7 | $8 \cdot 10^{-4}$ | 0.6k | 86M |
| ViT-L/16 | 14 | $4 \cdot 10^{-4}$ | 4k | 307M |
| ViT-H/14 | 14 | $3 \cdot 10^{-4}$ | 15k | 632M |
| ViT-H/14 - 4B | 1 | $3 \cdot 10^{-4}$ | 15k | 632M |
| ResNet-50x4 (BiT) | 7 | $1 \cdot 10^{-3}$ | 8k | 384M |
| ResNet-152x4 (BiT) | 7 | $6 \cdot 10^{-4}$ | 11.4k | 937M |

Table 6: Pre-training hyperparams. All models are trained on deduped JFT-300M with the exception of ViT-H/14-4B, which was trained on much larger JFT-4B dataset but for roughly the same number of steps as ViT-H/14. We used batch size of 4096, learning rate warmup of 10k steps and then linear decay. Additionally, we set dropout rate to 0.0, clip global norm to 1 and weight decay to 0.1. We use images of resolution 224x224.

**Training details.** All our models were pre-trained using TPUv4 hardware with exact amounts depending on the model. For reference, our smallest model ViT-B/16 took around 600 TPUv4 hours to pre-train and our biggest model ViT-H/14-4b took 25k TPUv4 hours.

## E  Finetuning details

We finetune on ImageNet train split and present the Top-1 accuracies we obtain from the official test split. Unless specified otherwise, we used images of input resolution 384x384 which is inception cropped (Szegedy et al., 2015) from a resolution of 448x448. In addition we perform horizontal flipping as data augmentation. These pre-processing steps exactly follow Kolesnikov et al. (2020); Dosovitskiy et al. (2021).

### E.1 SGD with Momentum hyperparameters

| Hyper parameters | Range |
|---|---|
| Learning rate | 0.03, 0.06, 0.1, 0.4, 1.6, 6.4, 25.6, 102.4 |

Table 7: Finetuning hyperparams. Models are trained for 10 epochs, with 0.25 epochs for warmup and cosine decay, no weight decay, no dropout and grad clipping at global norm 1. Similar to previous art, we fine-tune at a higher resolution of 384. When training the models with DP, we replace the global clipping with per example clipping norm of 1.0

### E.2 LAMB hyperparameters

| Hyper parameters | Range |
|---|---|
| Learning rate | 0.0001, 0.0005, 0.001, 0.005, 0.01, 0.05, 0.1, 0.5, 1.0 |
| Pre-trained layer LR multiplier $\alpha$ | 1.0, $10^{-1}$, $10^{-2}$, $10^{-3}$, $10^{-4}$, $10^{-5}$ |

Table 8: Finetuning hyperparams. Models are trained for 10 epochs, with 0.25 epochs for warmup and linear decay, no weight decay, no dropout and grad clipping at global norm 1. Similar to previous art, we fine-tune at a higher resolution of 384. When training the models with DP, we replace the global clipping with per example clipping norm of 1.0. In our earlier results we found that sometimes finetuning the full model performed much worse than finetuning just the last layer with privacy. We added an additional hyperparameter $\alpha$ which we multiply the global learning rate with for all layers except the last. The intuition is that since all layers except last have been pre-trained, the optimal choice of learning rate may be different for them compared to the last layer, which we start from scratch.

### E.3 Hyperparameters to Reproduce Figure 1b

In this setting, we only train the last layer for a single epoch in full batch setting. This amounts to a single update of the optimizer. We emphasize that we use the full dataset of around 1.28M training set as a single batch. This is different from using batch size of $2^{20}$ for 1 step, which leaves a subset of the images out. We initialize the last layer with zero and biases with -10 (default in Scenic). Additionally, we also changed the input resolution to 256x256 which is central cropped from an image of resolution 384x384. Finally, since we only make one update, we sweep over constant learning rate values $\in [10^{-4}, 10^{-3}]$ and make an update with DP-LAMB optimizer. Note that, for 1 step updates and zero initialization, LAMB is equivalent to Adam, thus we expect DP-Adam to produce similar results.

**Adam in single step full batch setting.** It is important to note that Adam in single step, full batch setting in fact reduces to signSGD where adam update $\frac{g}{\sqrt{g}+\varepsilon} \simeq sign(g)$. Sign based methods are indeed shown to be quite effective (Riedmiller & Braun, 1993) and in our case even surpass vanilla SGD. Since LAMB is equivalent to Adam when finetuning just one layer (with learning rate retuned), LAMB also reduces to signSGD in this restricted setting.

**Privacy accounting of hyperparameter tuning** We additionally note that we do not include any cost related to hyperparmeter tuning in our privacy accounting and budget. This is in line with the setup of the baselines we consider. Finally, since learning rate is the only hyperparamter we tune, at least in the last layer single step setting, we have found that results are quite robust to the exact value of the learning rate as long as it is low enough to not diverge.

# F   Pretraining with ImageNet21k

**Dataset.** ImageNet-21k is a superset of ImageNet-k with 21k classes and 14M images (Deng et al., 2009). Similar to before, in order to both not inflate our results and break privacy guarantee, we extend the deduplication process proposed by Kolesnikov et al. (2020) and deduplicate ImageNet-21k with respect to all splits of ImageNet-1k, CIFAR-10 and CIFAR-100.

**Pretraining Hyperparameters.** At the pre-training stage, we stick with the common practice of employing Adam optimizer (even for ResNet) with $\beta_1 = 0.9$ and $\beta_2 = 0.999$, with a batch size of 4096. Unlike pre-training with JFT dataset, we follow recommendations from Steiner et al. (2021) to use AugReg strategy where we lower the weight decay to 0.0001 and don't use dropout but instead use data augmentation strategy called **medium1** which combines Mixup with $\alpha = 0.2$ (Zhang et al., 2017) and RandAugment with $l = 15$ and $m = 2$ (Cubuk et al., 2020). We also use linear learning rate warmup until 10k steps and linearly decay it until the end. Our model is pre-trained with 224x224-sized images.

| Model | Epochs | Base LR | TPU v4 hours |
|-------|--------|---------|--------------|
| ViT-B/16 | 300 | $10^{-3}$ | 2.7k |

Table 9: Pre-training hyperparams. We used batch size of 4096, learning rate warmup of 10k steps and then linear decay. Additionally, we set dropout rate to 0.0, clip global norm to 1 and weight decay to 0.0001. We use images of resolution 224x224.

## F.1   Hyperparameters to Reproduce Table 5

Similar to Figure 1b, we only train the last layer for a single epoch in full batch setting. This amounts to just a single step of optimization. Similar to before, we initialize the last layer with zero and biases with -10 (default in Scenic). We also changed the input resolution to 256x256 which is central cropped from an image of resolution 384x384. This may look a little unusual at first for CIFAR-10 and CIFAR-100 since the original resolution of the images is 32x32. But we first upsample them to 384x384 and then central crop them. We found that using higher resolution images made a big difference in performance (even in non-private setting), especially when using features from a pre-trained model. Finally, since we only make one update, we sweep over constant learning rate values $\in [10^{-6}, 10^{-5}, 10^{-4}, 10^{-3}]$ and make an update with DP-ADAM optimizer as before.

**Privacy accounting of hyperparameter tuning** We additionally note that we do not include any cost related to hyperparmeter tuning in our privacy accounting and budget. This is in line with the setup of the baselines we consider. Finally, since learning rate is the only hyperparamter we tune, at least in the last layer single step setting, we have found that results are quite robust to the exact value of the learning rate as long as it is low enough to not diverge ($10^{-3}$ in this case).

# G   Regarding Reproducibility

In order to aid in reproducability of our work, we provide results with large open pre-training dataset called ImageNet-21k. In addition, we provide exact hyperparameters and other details in the Appendix. We will also release checkpoints and features from models pre-trained on ImageNet-21k. Finally, in order to help reproduce our results even further, we will also release our code.

# H   Supplementary Results

## H.1   Throughput comparison

|  |  | Tuning type | | |
|---|---|---|---|---|
| Model | Full | Last layer | Speed Up |
| ViT-B/16 | 170 | 900 | 5.3x |
| ViT-L/16 | 25 | 235 | 9.4x |
| ViT-H/14 | 12 | 100 | 8.3x |
| ResNet50x4 (BiT) | 25 | 225 | 9x |
| ResNet152x4 (BiT) | 3 | 90 | 30x |

Table 10: Throughput (img/sec/core) comparison of DP-LAMB on various models trained on Imagenet. All models were trained using 64 TPUv4 cores.

## H.2   More stringent delta

We conducted all our experiments with $\delta = $ 1e-6 in order for a fair comparison with Kurakin et al. (2022) but ImageNet dataset size is close to 1.28M. Here we re-did sweep over $\varepsilon$ on H/14-4b using the same hyperparameters used to obtain results in Figure 1b but with $\delta$ set to 8e-7. Note that De et al. (2022) also set $\delta$ to 8e-7. As shown in Figure 3, we find that the change in effective noise multiplier is small enough that our results don't change at all across both values of $\delta$.

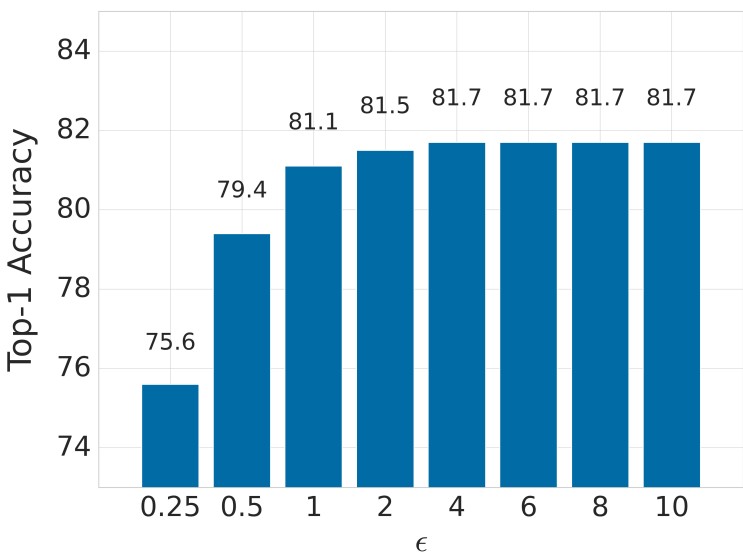

Figure 3: Comparison of Top-1 accuracies on ImageNet when varying privacy parameter $\varepsilon$ when $\delta$ set to 8e-7.

## H.3   ReaL labels

We also provide results when evaluated on ImageNet-ReaL (Beyer et al., 2020) labels in Figure 4.

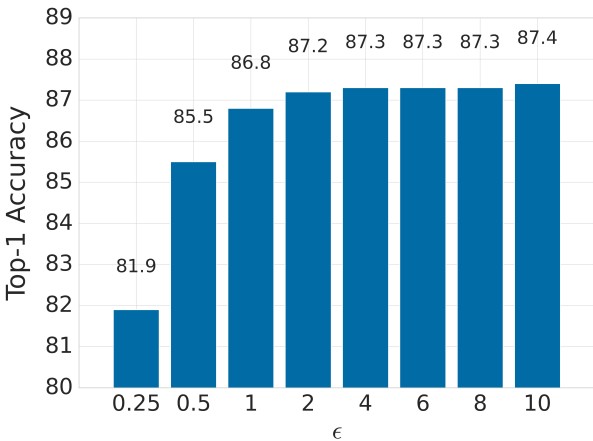

Figure 4: Comparison of Top-1 accuracies on ImageNet-ReaL labels (Beyer et al., 2020) when varying privacy parameter $\varepsilon$

| | | With privacy | |
| --- | --- | --- | --- |
| Model | Finetuning type | $2^{15}$ | $2^{16}$ |
| ViT-B/16 | Full | 45.23 | 58.6 |

Table 11: We complement results from Table 1 with full finetuning of ViT-B/16 on smaller batch sizes as well. As shown in Table 1 we obtain best results with batch size $2^{17}$.

## H.4   Smaller batch sizes

Since we get the highest accuracy with full finetuning of ViT-B/16 in Table 1 at the lower end of our batch size range, we also try batch sizes even lower than that to verify if that is indeed the maximum we could achieve. We verify this as shown in Table 11.

