# OpenReview forum: "Towards Large Scale Transfer Learning for Differentially Private Image Classification"
_TMLR — Accepted by TMLR_

### Review · Reviewer_paed · 2022-11-25

**Summary Of Contributions:**

This paper presents an experimental study of large scale transform learning for differentially private image classification. Authors study the effects of model-architecture and optimization methods in DP fine-tuning of a large image classification deep learning model. Based on their experimental results, authors suggest three practices for the DP transfer learning for this particular set of models:
1. "Increase pretraining data and model size.". Authors show in their experiments (Table 3), that using a larger pre-training data set improves the performance of both non-private and private classification. Authors also show that using a more complex model improves the fine-tuning performance for both private and non-private cases.
2. "Use large batch sizes in combination with optimizers that can work well with large batch sizes (e.g.
LAMB).". Motivated by prior work on the topic, authors experiment with large batch sizes in the DP fine-tuning. The experiments suggest that using a large batch size and training for only few iterations improves the DP performance. Towards the end of the paper authors suggest using only a single gradient update step with the full data set used as the "batch".
3. "Initialize the last layer to zero (or very small) when doing finetuning with DP.". In their experiments, authors demonstrate that fine-tuning only the last layer of the classification model provides on par performance to the full fine-tuning when trained under DP and using a large batch size. In their ablation study authors show that initializing the fine-tuning with zeros (the weights of the last layer) provides significant performance increase compared to the random initialization.

**Audience:**

Yes

**Broader Impact Concerns:**

I have no concerns relating to the broader impact.

**Claims And Evidence:**

No

**Requested Changes:**

I require following changes:
- The Adam should be better discussed to understand the results of the single step training.
- The models should be better described in the appendix, including for example a clear description of the model size.

More minor changes that will improve the paper
- The captions for the Tables should be more descriptive to convey the main message of the Table.

**Strengths And Weaknesses:**

The empirical results shown in the paper are very strong and improve over the state-of-the-art significantly. While the work does not present theoretical reasoning for the proposed improvements for the learning procedure, I find this type of empirical work also valuable for building actually functioning DP algorithms. Also, I think the topic is relevant and of high interest to the TMLR community. However, I do have some serious concerns about the method that need to be addressed before the empirical performance can actually be qualified.

If I do understand the paper correctly, the results are based on fine-tuning the model for only a very iterations. The ablation study in Table 4 shows that the performance is actually the best if we train only for a single epoch. Moreover, the authors suggest that the extreme case of training only a _single step_ using all the data as the "batch" provides the best performance. Now, if we use the Adam optimizer from Kingma et al. (I assume that the DP-Adam is just a DP variant of this where the noise is added to the clipped gradients prior to learning rate adaptation), then the first gradient update step should actually correspond to a constant update. This is easy to see from the description of the Adam in the original paper. The moment parameters of Adam are initialized as $m_0=0$, $v_0=0$ and the first update becomes:
\begin{align}
   &m_1 = (1-\beta_1) g \\\\
   &v_1 = (1-\beta_2) g^2 \\\\
   &\hat{m}_1 = m_1 / (1-\beta_1) = g \\\\
   &\hat{v}_1 = v_1 / (1-\beta_2) = g^2 \\\\
   &\theta_1 = \theta_0 - \alpha \hat{m}_1 / (\sqrt{\hat{v}_1} + \epsilon) = \theta_0  - \alpha g/(g + \epsilon)
\end{align}
So if the $\epsilon$ (note that this is not the privacy parameter) in the denominator is chosen significantly smaller than the gradient (which typically is the case), then the first step in Adam becomes essentially $\theta_1 =  \theta_0  - \alpha$ and the update is independent of the gradient. Therefore the model update happens independently of the data. The results for the CIFAR-10 and CIFAR-100 seem to be almost non-affected by the epsilon, which could support the hypothesis that the gradient step actually does not use the data at all. However, the results for the ImageNet-1k of Table 5 seem to be affected by the privacy parameter $\epsilon$, which would suggest that the method is actually dependent on the data. Can authors clarify the use of Adam in the single iteration setting? If the DP-Adam is somehow fundamentally different to the Adam, please specify it in the appendix.

Besides the possible problem with single step learning pointed above, I want to point out some other weaknesses of the paper as well.
- Simply using more data for pre-training seems somewhat unattainable suggestion for many cases. If there is a large public data set available that could be use to pre-train the model, then in what scenario the analyst _wouldn't_ use it? I think this is crucial point to consider for better motivating the suggestion 1.

- Authors should better specify the model architectures, and which models are more complex than other. I didn't find much discussion about this in the paper. For example I don't quite follow the discussion after Table 3. You state that the accuracy gap decreases with larger models. And then you state "For instance, the gap for ViT-L/16 (last layer tuning) between the best non-private and private case is 7.5 percentage points and for ViT-H/14 in the same setting is 6.2 percentage points.". So is the ViT-L/16 model smaller than the ViT-H/14?

Some minor comments:
- Captions of Table 2 and 3 are equivalent. While the Tables might contain overlapping settings, the captions should at least describe the main result of the table and the particular effect studied.
- Section 5.1: What do you mean by "orthogonal" here? Orthogonal compared to what?
- Figure 1a is not referred in the main text.
- Caption of Figure 1a says "across model and pre-training dataset sizes.", but it remained bit unclear to me what are the data sets and models corresponding to the labels in the x-axis.
- The appendix says that the code is included but I didn't find it from the openreview system. Maybe it was there, but just didn't show for me.

Typos:
- "computeing"

---

> ### Author Response · Authors · 2022-11-30
> **Response**
>
> We thank the reviewer for the review and encouraging comments! In addition to addressing the main concern, we have incorporated all the feedback from the reviewer in the attached revision. We hope that our response alleviates your concerns.
>
> ```Response for main concern related to Adam in single step setting```
>
> Thank you for pointing out the calculation of Adam for a single step. We were also a little surprised at the effectiveness of Adam even in the single step setting.
>
> Your calculation correctly observes that $\theta_1$ has only a minor dependence on the norm of the gradient vector $g$. However, you may have overlooked the fact that $\theta_1$ does in fact have a strong dependence on the *direction* of $g_t$ (given by the sign of each coordinate) because $\sqrt{v}\approx |g|$ rather than $g$. Since the direction of $g$ depends on the data, the overall computation of $\theta_1$ does indeed depend on the training data set. In the single step adam setting, the update simplifies (roughly) to signSGD where $\text{update} = \frac{g}{\sqrt{g}} = \text{sign}(g)$. Outside the context of DP, signSGD has in fact been shown to be quite competitive (https://arxiv.org/abs/1802.04434) and in our case even outperform vanilla SGD in large batch setting. We have added more discussion on this aspect of Adam/LAMB in the paper in appendix section E3
>
> Now to see whether DP changes anything, modulo clipping operation, the noise that gets added to $g$ to preserve DP is scaled roughly as $\mathcal{N}(0,\frac{L\sqrt{\log(1/\delta))}}{\epslion n})$, where L is the $\ell_2$-Lipschitz constant of the individual gradients and $n$ is the training data set size (as we are operating over full-batch). The scale of the noise that gets added to each coordinate can be much smaller than the overall scale scale of the gradient $g$, if the number of training examples $n$ is large. Hence, the signs of the gradient g are sufficiently preserved under DP..
>
> ```Authors should better specify the model architectures..```
>
> You are absolutely right, we have added model size information in Table 6 and the details of the architecture can be found in appendix section C1.
>
> ```Captions of Table 2 and 3 are equivalent.```
>
> Thanks for pointing that out, we have updated the caption for Table 3 to summarize the results.
>
> ```The appendix says that the code is included..```
>
> Apologies for any confusion, we are working on releasing our code for private finetuning but it may take a few more weeks. However, currently, we have tried our best to include all details regarding the training procedure (both pretraining and finetuning), we are also more than happy to address any questions you may have in the meantime.
>
> ```Simply using more data for pre-training seems somewhat unattainable suggestion for many cases..```
>
> We totally understand your concern! In our recommendations and our study, our hope is to illustrate the effectiveness of a large public pre-training dataset when it's actually available. Before our work, it was not a priori clear that a simple strategy of large pre-training dataset + large model would be an effective strategy to train high-performance DP models that approach even non-DP benchmarks. Even if for some use cases, a large pre-training dataset is not available or is not in-distribution to the sensitive dataset that we want privacy over, our work provides a lower bound on the utility and value of gathering such in-domain public data.
>
> Having said that, as you pointed out, we completely agree that this strategy is not a panacea and it deliberately leaves out use cases where transfer learning is not easy to apply (for instance, recommendation systems or click through rate prediction where features are sparse and opaque).

---

> > ### Comment · Action_Editors · 2022-11-30
> > **Code release**
> >
> > Is there any estimate on when the code will be ready for release? If it's claimed as a contribution of the work, it seems reasonable that reviewers should have the option to see and evaluate it.

---

> > > ### Author Response · Authors · 2022-11-30
> > > **Response to code release**
> > >
> > > That's totally fair! However, we do not claim code release as a contribution of the work (we mention that we **will** release it in the appendix but not that its immediately available with the submission) but we are more than happy to do it in a few weeks (right now a few stakeholders are on vacation which is contributing to the delay). Finally, definitely let us know if you had any specific concerns!

---

> > ### Comment · Reviewer_paed · 2022-12-05
> > **After rebuttal**
> >
> > Thanks for addressing my concerns! I see now, that the single step Adam indeed corresponds to the sign-SGD. I'm also satisfied with the reasoning for the small changes caused by DP. I would encourage the authors to add brief discussion about the changes observed in the ImageNet-1k experiment. I guess what happens there, is that the gradients are sufficiently small around the initialization point, and the DP noise (especially for the $\epsilon=0.5$) is able to flip the signs of the gradients which leads to decrease in accuracy.
> >
> > One more thing that I would like to be clarified is the hyperparameter selection setting. Am I correct that the hyperparameter selection is not being accounted in the privacy budget? This is not a huge deal, as it is rarely taken into account in many DP-SGD applications. However, since the sign-SGD will completely omit the strength (the magnitude) of the gradient signal, I would imagine the learning rate plays a big role in the fine-tuned model's performance. In appendix, you say that you sweep over four learning rate values to reproduce Table 5. I think it would be appropriate to discuss the effect of this parameter tuning and mention that the DP cost is not accounted for. If I'm mistaken, and you have taken this already in account, please disregard my comment.

---

> > > ### Author Response · Authors · 2022-12-07
> > > **Response to after rebuttal**
> > >
> > > We are glad that we were able to alleviate some of your concerns! We will try to further address additional concern you raised below.
> > >
> > > ```One more thing that I would like to be clarified is the hyperparameter selection setting..```
> > >
> > > It is a fair point that we did not account for the privacy cost for hyperparameter tuning for this paper, which is consistent with all the baseline results we discussed. We will explicitly mention this fact in the paper in the appendix with our main results, and also some additional discussion for the hyperparameter choices mentioned in the paper. Additionally, since learning rate is the only hyperparamter we tune, at least in the last layer single step setting, we have found that results are quite robust to the exact value of the learning rate as long as it is low enough to not diverge. This can further help reduce the privacy cost of hyperparmeter tuning with our recommendations (even lower than cost of sweeping over 4 LRs we currently do). As an aside: Since the number of choices of the learning rate is not too large, in principle one can use https://arxiv.org/abs/2110.03620 to do that with DP. We will mention about this in the conclusion section.
> > >
> > > Let us know if you had more questions!

---

### Review · Reviewer_GzUe · 2022-11-26

**Summary Of Contributions:**

The paper studies the application of transfer learning through pre-training for image classification, in a private setup. The paper shows that privately  fine-tuning pre-trained models helps close the gap between private and non-private setups. Based on their empirical analysis, the authors make recommendations on what best practices are when using DP-SGD for image classification. They show that using the right optimizer and initialization for the classification layer are both very important in achieving desirable privacy-utility trade-off. They also show that through their recommendations, the training time of models can be severely decreased.

**Audience:**

Yes

**Broader Impact Concerns:**

The paper facilitates wider deployment of private training algorithms which in itself can have many positive impacts. However, my only ethical concern is with the training time and hardware, as it seems training takes in the order of thousands of hours, using TPU v4s (based on Appendix D). Given how pre-training is what actually causes the improvements shown in the paper, I think it would be beneficial to the community if the pre-trained models are made publicly available to help accessibility, and alleviate the need for re-training them over and over by different parties.

**Claims And Evidence:**

Yes

**Requested Changes:**

My main question/potential request for more experiments is regarding the initialization of the last layer, for fine-tuning. right now zero initialization and Lacuna initialization are studied, and there seems to be a huge gap between their performances. I wonder if the authors tried other initializations? Or even maybe a way to initialize with transfer learning and using parameters from either other layers or other models?

Another question I had was regarding Tables 3 and 4: The performance of the best setup (the final row in table 3) which is the highest scaled one seems to be invariant of optimizer choice (according to table 4). Do the authors think that this might not be the case if full-fine tuning is performed? Or could it be that the large scale of the data/training kind of overshadows the effect of the optimizer? (I mainly mean between Adam and LAMB).


Some minor comments/suggestions:

1. Page 1: Introduction, end of line 3, differentially privacy should probably be differential privacy
2. Page 2: Introduction, line 5, might significant increase -> might significantly increase
3. Page 8: end of last line "that that" -> that




**Strengths And Weaknesses:**

Strengths:

1. I really liked how the authors made sure to deduplicate the pre-training data from the fine-tuning data, that is a very important step which is missing in many similar related work, and makes the conclusions much more reliable.

2. The paper's narrative is very easy to follow, and I enjoyed how it answered every question that came to my mind in the next paragraph. The results are thoroughly explained and justified.

3. Studying the effect of the optimizer on performance, and also analyzing the computational overheads (run-time) and achieving faster run time are aspects I had not seen studied much before in similar related work.


Weakness:

1. Although there are aspects to this paper that differentiate it from prior work, the main idea is still using pre-training which has been explored before multiple times. However, I don't think this is really that much of a weakness as the findings and insights do have novelty.

---

> ### Author Response · Authors · 2022-11-30
> **Response**
>
> Thank you for your comments and careful consideration! We hope that our response alleviates your concerns!
>
> ```My main question/potential request for more experiments is regarding the initialization of the last layer, for fine-tuning..```
>
> Indeed, in our initial exploration, we had tried different initialization schemes like Lecun, He Kaiming and Glorot init. All of these can be instantiated as random normal (or uniform) distribution with a change only in the variance parameter. Thus, at least in the setting of training just the last layer, the only thing which is affected by choosing one init scheme over the other is the scale of the parameters. Given this, we decided to directly sweep over the scale parameter as shown in Figure 2(c). We observe that there is nothing special about initializing at zero and as long as the scale is small enough, we can recover the performance at zero init.
>
> Regarding "initializing with transfer learning and using parameters from either other layers or other models": this is a very interesting idea which we hadn't considered at all. Perhaps it may be possible to slice the columns of the last layer of pre-trained weights for classes which were common across pre-training and fine-tuning time. Is that what you had in mind?
>
> ```Another question I had was regarding Tables 3 and 4..```
>
> That is a good question! So, the way we were thinking about it was that, at least in the last layer finetuning setting, DP-LAMB in fact reduces DP-Adam since the additional multiplicative factor of $||w|| / ||g||$ in LAMB can be subsumed in the learning rate. Thus the lack of difference in performance between DP-Adam and DP-LAMB in Table 4 may be explained this way.
>
> On the other hand, for full finetuning, it is unclear whether DP-LAMB and DP-Adam would result in similar performance. However, looking at Table 3, the difference in performance between full finetuning vs last layer tuning at 2^20 batch size seems to diminish as the model/dataset is scaled to the point where full finetuning of ViT-H/14-4b results in the same accuracy as finetuning just the last layer, even in the non-private setting (82.5%). This suggests that, even if DP-Adam were used instead of DP-LAMB, after proper tuning of the learning rate, one may be able to recover the same accuracy since there is at least one configuration of learning rate where it can get the same accuracy as DP-LAMB (i.e. learning rate of 0 for all the layers except last layer). Does this make sense? Let us know if you had any further questions.
>
> ```ethical concern is with the training time and hardware, as it seems training takes in the order of thousands of hours```
>
> You are absolutely correct and we share your concern! We would like to note that, we only had to retrain at pre-training time since we had to deduplicate the pre-training and finetuning datasets. Without this requirement, we would have been able to repurpose pre-trained checkpoints. More recently though, we have observed that this deduplication step has become a standard practice even in the non-private setting (https://arxiv.org/abs/2106.04560, https://arxiv.org/abs/2205.01917) and off-the-shelf pre-trained models can in fact be repurposed for DP purposes as well. In fact, several high performance pre-trained vision transformer models can be found at https://github.com/google-research/vision_transformer#available-vit-models.
>
> Finally, our results suggest that, after sufficient scaling of the model/data, privately finetuning just the last layer is sufficient to train a high-performance DP model. Given the computational challenges and cost of training the full model with DP-SGD, we argue that our results ultimately point towards amortizing and increasingly leveraging already trained models for high-performance DP training, and thus potentially substantially **reducing** the overall energy consumption instead of increasing it.

---

### Review · Reviewer_UVNH · 2022-11-28

**Summary Of Contributions:**

The paper describes large-scale experiments with transfer learning and DP-SGD. The results are very important for our community as they demonstrate that at large-scale the model performance can reach non-private baselines.

**Audience:**

Yes

**Broader Impact Concerns:**

I would try to still say that public data does not always mean that the model can be created or that data can shared. I think your examples with JFT emphasizes this distinction, which is a very good argument for me.

**Claims And Evidence:**

Yes

**Requested Changes:**

I think that the paper might get stronger with a discussion on the difference between private and public data.

My biggest concern is the experimental setup: a public ImageNet dataset is considered **private** and the private JFT dataset is considered **public**. The described approach sounds to me practical, however I wonder what is the meaning of the "public data" that is not publicly available. Is it reasonable to expect that datasets like JFT will become publicly available or due to sensitivity of large sets of data it might never happen? I also might suspect that other domains, e.g. text, might have more data publicly available.

I hope this request does not sound too abstract, however as you describe benefits of using public data for private training, I feel like it's important to dedicate a section on "defining" the public and private data. For example, using privacy frameworks like contextual integrity to emphasize that even publicly data might still not allow to train models for certain cases or be released in an aggregated form.

**Strengths And Weaknesses:**

- The paper has a very good experimental setup and set of results that are very convincing.
- A thorough description of the DP-SGD


Weaknesses:
- More discussion on private vs public (see "requested changes" section)
- Emphasize other domains, e.g. text, as possible other domains where this method might be relevant
- Also mention other settings, e.g. federated learning, as possible benefits for training large models.

---

> ### Author Response · Authors · 2022-11-30
> **Response**
>
> We thank the reviewer for the review and encouraging comments. We hope that our following response to the **public data question**  alleviates your concern:
>
>
>
> ```Regarding public and private data```
>
> Thanks for bringing up this important aspect of our work, and in general of  DP-Finetuning in the presence of public data. First off, while it is true that the JFT data sets are not available publicly, we did use deduplication techniques by Kolesnikov et al. 2020 to ensure that JFT and Imagenet do not have overlap.
>
> Second, we fully agree with you that the terminology is a bit confusing. We differentiate "public data" and "private data" only in the context of privacy. We definitely do not imply that models trained using "public data" (in dp sense) can necessarily be released or can eschew scrutiny. We had tried to clarify this bit in Section 4, and have expanded/emphasized that discussion more in our revision as requested.
>
> Finally, the main point of this work is to empirically demonstrate that one can indeed train on datasets with similar complexity as that of Imagenet with strong DP guarantees, if one is able to pretrain on a reasonable data set (which can or can not be shared). It was not a priori clear but well known by now that pre-training avoids a lot of the challenges due to the curse of dimensionality in DP training (see https://www.lxuechen.com/publication/dp_low_rank/).

---

> > ### Comment · Reviewer_UVNH · 2022-12-04
> > **reponse**
> >
> > thanks, I like the clarification and think that it makes the discussion better. I don't have any more concerns.

---

> > > ### Author Response · Authors · 2022-12-07
> > > **Thank you**
> > >
> > > We are glad that we were able to alleviate your concerns, thank you for your time and effort!

---

### Comment · Action_Editors · 2022-11-28
**Revision and Discussion Period**

Reviewers seem generally positive about the paper. They have a few technical questions, requests for additional experiments, and ask for more discussion about the nature of public and private data in this setting.

Note that formal decisions will be submittable in two weeks. The authors should try to address these comments and update the manuscript (ideally marked in another colour) before then.

---

> ### Author Response · Authors · 2022-11-30
> **Thank you!**
>
> We have posted official response to all the reviewers and uploaded a revision with suggested changes. Thank you for your consideration!

---

### Decision · Action_Editors · 2023-01-02

**Recommendation:** Accept with minor revision

**Comment:**

The reviewers all thought the work was great, and were generally satisfied by the revisions by the authors. I too agree: I think this is a really nice work, which pushes the limits of public pre-training, showing that differentially private fine-tuning can preserve most of the utility. Congratulations to the authors.

For the paper itself, I would "Accept as is." However, as the authors have promised, the code is in the process of being publicly released. The only remaining concern of the reviewers (and myself) is that it has not yet been released. To add some impetus to this process, I am marking the paper as "Accept with minor revision." To satisfy the minor revision, I would like either a) the code to be publicly released and added to the submission, or b) the authors to publicly commit to a timeline for code release (in the case that administrative overhead precludes releasing within this limited timeframe).

**Audience:**

This paper is of high interest to anyone working in differentially private machine learning, which I consider to be a large community within TMLR's audience.

**Claims And Evidence:**

The main claim in this paper is effectively that pretraining (publicly) on large-scale datasets can significantly improve the utility of downstream (differentially private) fine tuning. This claim is thoroughly supported with a number of experiments.

---

> ### Author Response · Authors · 2023-01-05
> **Thank you**
>
> We would like to thank the action editor and the reviewers! Your feedback has been quite valuable and has made the paper stronger. We are glad to see our work accepted at TMLR and grateful for the kind words.
>
> Regarding the code: we deeply appreciate your patience on the matter. We are now very close to making the code public, which would help reproduce results from our recommendations. Most likely, we should be able to do it in next 2-3 weeks, if not sooner. We hope that this helps alleviate your concern.

---

> > ### Comment · Action_Editors · 2023-01-15
> > **Thanks**
> >
> > As discussed via email, we'll be holding off on approving the currently submitted CR, and the authors are making an effort to get the code online by the end of the month.